# "How is your thesis going?"–Ph.D. students' perspectives on mental health and stress in academia

**Julian Friedrich**[1,2]\*, **Anna Bareis**[1,2], **Moritz Bross**[1,2], **Zoé Bürger**[1,2], **Álvaro Cortés Rodríguez**[1,2], **Nina Effenberger**[1,2], **Markus Kleinhansl**[1,2], **Fabienne Kremer**[1,2], **Cornelius Schröder**[1,2]

**1** Eberhard Karls Universität Tübingen, Tübingen, Germany, **2** *sustainAbility* Ph.D. Initiative at the Eberhard Karls Universität, Tübingen, Germany

\* julian.friedrich@uni-tuebingen.de

**Data Availability Statement:** The anonymized data set is available at https://doi.org/10.23668/psycharchives.12914. All code for the analysis can

## Abstract

Mental health issues among Ph.D. students are prevalent and on the rise, with multiple studies showing that Ph.D. students are more likely to experience symptoms of mental health-related issues than the general population. However, the data is still sparse. This study aims to investigate the mental health of 589 Ph.D. students at a public university in Germany using a mixed quantitative and qualitative approach. We administered a web-based self-report questionnaire to gather data on the mental health status, investigated mental illnesses such as depression and anxiety, and potential areas for improvement of the mental health and well-being of Ph.D. students. Our results revealed that one-third of the participants were above the cut-off for depression and that factors such as perceived stress and self-doubt were prominent predictors of the mental health status of Ph.D. students. Additionally, we found job insecurity and low job satisfaction to be predictors of stress and anxiety. Many participants in our study reported working more than full-time while being employed part-time. Importantly, deficient supervision was found to have a negative effect on Ph.D. students' mental health. The study's results are in line with those of earlier investigations of mental health in academia, which likewise reveal significant levels of depression and anxiety among Ph.D. students. Overall, the findings provide a greater knowledge of the underlying reasons and potential interventions required for advancing the mental health problems experienced by Ph.D. students. The results of this research can guide the development of effective strategies to support the mental health of Ph.D. students.

## Introduction

Work situations can be demanding and have a profound influence on employees' mental health and well-being across different sectors and disciplines [1]. Multiple studies show that the mental health status of people working in academia and especially that of Ph.D. students seems to be particularly detrimental when compared to the public [e.g., 2,3]. Disorders such as

be found at https://github.com/coschroeder/
mental_health_analysis.

**Funding:** We acknowledge support by the Open Access Publishing Fund of University of Tübingen. The funders had no role in study design, data collection and analysis, decision to publish, or preparation of the manuscript.

**Competing interests:** The authors have declared that no competing interests exist.

anxiety and depression are on the rise in the general population [4,5]. Multiple studies show that this is even more severe in academia [6–10] and in particular Ph.D. students are affected by mental health problems [11,12]. Worldwide surveys grant support for Ph.D. students' sub-optimal and alarming mental health situations [13,14].

A comprehensive study with more than 2000 participants (90% Ph.D. students, 10% Master students) from over 200 institutions across different countries showed that graduate students were more than six times more likely to experience symptoms of depression and anxiety than the general public [2]. Furthermore, a global-scale meta-analysis [3] and several other studies concerned with the mental health of Ph.D. students in different countries, e.g., the United States [7,9], the United Kingdom [6], France [15], Poland [8], Belgium [16] or Germany [11,12] voice concerns about the mental health situation of Ph.D. students. Recent research conducted in Belgium has consistently found a higher prevalence of mental health problems among Ph.D. students compared to different groups of other highly educated individuals [16]. In the same study, 50% of the Ph.D. students reported that they suffer from some form of mental health problem, and every third is at risk of a common psychiatric disorder [16]. A similar picture is forming in Germany. For example, the prevalence of at least moderate depression among doctoral researchers at the Max Planck Society, one of the biggest academic societies in Germany, was between 9.6% and 11.6% higher than in the age-related general population [11].

## Increasing numbers of anxiety and depression among Ph.D. students

Recent studies describe not only a high prevalence but also a rising tendency of mental health issues among Ph.D. students. In a study from 2017, 12% of the respondents reported seeking help for depression or anxiety related to their Ph.D. [13], while in 2019, the result was even more drastic, as 36% of the respondents reported that having searched for help for those same reasons [14]. Several studies among doctoral researchers within the Max Planck Society show similar results. For instance, a survey in 2019 showed that the average of the Ph.D. students were at risk for an anxiety disorder and another sample from 2020 provided even more robust support for this claim [11,12]. Furthermore, the mean depression score increased from 2019 to 2020 in both samples [11].

## Risk factors and resources

Given these alarming statistics, several studies addressed risks and resources for increased mental health issues. Other studies have revealed that gender, perceived work-life balance, and mentorship quality are correlated with mental health issues [2,17]. Specifically, female gender [17] and transgender/gender-nonconforming Ph.D. students are, on average, more likely to suffer from mental health issues [2]. In contrast, a positive and supportive mentoring relationship or a supervisor's leadership style, and a good work-life balance are positively associated with better mental health [2,16]. While some authors [18] reported a negative correlation between the Ph.D. stage and mental health, with students at later stages disclosing greater levels of distress, others [16] did not find significant differences in this regard. Moreover, another report identified that Ph.D. students' satisfaction levels strongly correlate with their relationship with their supervisors, number of publications, hours worked, and received guidance from advisors [19]. Furthermore, several studies showed a positive correlation between job satisfaction [20,21] as well as a negative correlation between job insecurity [22] and mental health or perceived stress, also in Ph.D. students.

## Aim and research questions

Taken together, the alarming findings on the psychological status of Ph.D. students around the globe cannot be denied. However, data on the situation of Ph.D. students in Germany are scarce [11,12,23]; thus, comparisons of different universities within a country can hardly be made. However, addressing those differences is particularly relevant since the working conditions, concerning contract types, financial situations or supervision vary strongly among different countries, geographical regions and universities or institutions [24]. Furthermore, little is known about the reasons for this precarious situation and where exactly the need for action lies [25]. Therefore, the aim of this study was to conduct a survey among Ph.D. students at a university in the southwest of Germany to assess Ph.D. students' mental health status. Additionally, the present study also reveals information on the extent of the need for additional support services and pinpoints the specific areas where these services ought to be emphasized. In order to help identify relevant indicators, this investigation provides empirically sound findings on the mental health situation of Ph.D. students in Germany.

## Materials and methods

### Sample and procedure

Overall, 589 participants (60.3% female, 0.8% of diverse gender, $M_{Age}$ = 28.8, $SD_{Age}$ = 3.48, range 17–48 years) out of a total of enrolled 2552 Ph.D. students (response rate: 23.1%; actual numbers of Ph.D. students at the University of Tübingen higher as some Ph.D. students are not enrolled) took part in an online survey from October to December 2021. Instructions, items, and scales were all presented in English. Participants could answer the open questions in German or English and were comprised of Ph.D. students across various stages of their Ph.D. at the University of Tübingen without further exclusion criteria. The online questionnaire was sent to Ph.D. students' email addresses via mailing distribution lists in cooperation with the central institution for strategic researcher development (Graduate Academy) of the University of Tübingen and with Ph.D. representatives of different faculties. Ethics approval was obtained by the "Ethics Committee of the Faculty of Economics and Social Science of the University of Tübingen" and written informed consent was given by the participants.

The distribution of faculty affiliation of the participants was heterogeneous with shares of 61.8% Science, 12.4% Humanities, 11.7% Economics and Social Sciences. These numbers reflect the different sizes of faculties and are roughly aligned with the relative numbers of students (41.7% Science, 24.8% Medicine, 16.2% Humanities, 7.5% Economics and Social Sciences), with a clear underrepresentation of the Medical Faculty. Faculties with less than 20 participants or participants with multiple answers were grouped into one category for further analysis (Others 14.1%, see S1 Table). 67.9% of the participants were German and in total, 82.9% came from European countries. During data collection, the participants were at different stages of their Ph.D. ranging from 0 to over 130 months with a mean time of two and a half years (30.0 months) of Ph.D. progress.

### Measures

First, demographic data and background information on the current Ph.D. situation were collected. In a second part, to get a differentiated view, we included different measures to operationalize the mental health status of Ph.D. students. The quantitative questionnaire assessed 1) general health, generalized anxiety disorder, as well as internally reviewed self-generated questions, 2) life and job satisfaction, and quantitative job insecurity, and 3) stressors (institutional and systemic), causes of stress and potential solutions. This study also collected information

regarding the degree of participants' familiarity with the mental health resources available at the university, e.g., points of contacts for counseling, in order to evaluate whether Ph.D. students make use of these services. Moreover, participants were asked to name additional services that they may consider necessary.

**General health and stressors.** General health was assessed by two items of the Perceived Health Questionnaire (PHQ-2) [26]. Participants were asked to indicate how frequently they had experienced depressed moods and anhedonia over the past four weeks on a scale from 1 (not at all) to 4 (nearly every day). Additionally, they were presented with seven items of the Generalized Anxiety Disorder scale (GAD-7) [27] capturing the severity of various anxiety signs like nervousness, restlessness, and easy irritation on a scale from 1 (not at all) to 4 (nearly every day). Both scales were used in this combination in a previous study in German higher education [28]. Furthermore, we included two binary questions on whether the participants are currently in psychotherapy and if they have ever been diagnosed with a mental disorder.

The condensed version of the Perceived Stress Scale (PSS) [29] was used to get the degree of stressful situations in life in the last twelve months or since the start of the Ph.D. [30]. The response scale ranged from 0 (never) to 4 (very often), the following being a sample item: "... how often have you felt that you were unable to control the important things in your life?" To check the internal consistency of the four items, we calculated Cronbach's alpha which was .79.

**Job satisfaction and life satisfaction.** Three items on a scale from 1 (strongly disagree) to 5 (strongly agree) were used to measure job satisfaction [31], where a higher mean score indicated higher job satisfaction. A sample item is: "I am satisfied with my job." Cronbach's alpha was .86. Additionally, we added one item concerning general life satisfaction [adapted from 32] with the same response categories to get a more holistic insight.

**Job insecurity.** To assess the fear of losing the job itself, quantitative job insecurity was measured with three items (e.g., "I am worried about having to leave my job before I would like to.") [33] on a scale from 1 (strongly disagree) to 5 (strongly agree). We calculated a mean score with higher scores indicating higher job insecurity. Cronbach's alpha was .80.

**Institutional and systemic stressors.** For institutional stressors, we focused mainly on the role of supervision and included eight questions, four were framed using positive wording and four with negative wording, each with a scale from 1 (not at all) to 5 (all of the time). We summarized these questions in two constructs (positive support/negative support) which had Cronbach's alphas of .85 and .76, respectively. As for systemic stressors, we included two questions on long-term contracts and on future perspectives, again using a scale from 1 (strongly disagree) to 5 (strongly agree).

**COVID-19.** To cover the potential impacts of the COVID-19 pandemic and the implemented regulations, we included two questions to evaluate whether the pandemic affected the students' general situation. On the one hand, participants were asked to pick the statement that best describes the effects of the pandemic in general ("yes, it improved my general situation", "yes, it worsened my general situation", "yes, but it neither worsened nor improved my general situation", "no"), and on the other hand, they were asked to evaluate whether the particular answers provided in this survey had been affected by the pandemic from 1 (very likely) to 5 (very unlikely).

## Rating procedure and open answers

**Causes of stress and potential solutions.** We included three open-ended questions in the questionnaire to get a deeper understanding of the perceived causes of stress, potential ways to improve mental health, and ways to improve the overall situation of Ph.D. students. The questions were: (1) "What is/are the cause(s) of your stress?" (2) "What would need to change to

improve your mental health status?" (3) "What could be done to improve your situation?" Participants could mention as many points as they wanted (without any word limit). To analyze these questions, we built categories by following the model of inductive category development [34]. Two raters screened the first and last 20 responses in the data set and created categories for reoccurring topics (for a list containing all categories see S5–S7 Tables). In the next steps, two new raters rated all open answers with the developed categories and added additional categories if needed. Applicable categories were rated with 1 ("category was mentioned") or 0 ("category was not mentioned"). For example, the following response to question (1) "[My] supervisor is on maternity leave with open end, i.e. I have no one to talk to about my topic and have almost nothing so far [. . .] I feel like I'm not good enough at this, not sure I will be able to succeed–everyone else has other projects and publications except me–no topic-related network" was rated with 1 in the following four categories: supervision (quality & quantity), social integration & interactions (private & professional), self-perception (internal factors), and perceived lack of relevant competences & experience–(sense) of progress and success. The full list of categories and inter-rater reliability as measured by Krippendorf's Alpha is reported in Table 3 [35].

## Results

### Descriptive statistics of work environment and workload

The largest part of the participants (65.5%) was temporarily employed, 12.1% got a scholarship, 7.6% were permanently employed, and 6.5% were not employed at all. The mean for total contract length was 34.3 months, with a range between two and 72 months. About 10.5% of the participants had a contract for only 12 months or shorter. A similar large variation was found in the percentage of employment with a mean of 63%, ranging from 10% to 100% of employment. For workload, we found a mean of 36.0 hours of Ph.D.-related work per week with a standard deviation of 15.6 hours. After taking a closer look at high workloads, we found that 31.3% of the participants work 45 hours or more (21.5% work 50 hours and more) per week. On top of their Ph.D. work, many Ph.D. students work in other jobs, which combined with the hours spent for Ph.D.-related work, summed up to the mean of 44.1 overall working hours per week. A detailed description can be found in S1 Table.

### Faculty-wise comparison

In an explorative manner, we compared the mean differences of the most important variables between different faculties. Most of the analyzed variables did not show significant differences. Still, we want to stress that the highly imbalanced sample sizes (see S3 Table) could lead to false negative outcomes due to the small numbers of participants in some groups. However, we found that the mean job insecurity was significantly different between faculties ($p < .001$, Kruskal-Wallis rank sum test) with comparable low job insecurity in the faculties of law ($M = 2.10$, $SD = 1.22$) and theology ($M = 2.38$, $SD = 1.19$) and high insecurity in the faculty of humanities ($M = 3.32$, $SD = 0.91$).

### COVID-19

In total, 41.9% of the participants stated that their general situation worsened due to the pandemic, while 28.5% stated that the pandemic affected but it neither worsened nor improved their general situation. 33.5% of the participants stated that their responses in this study were "very likely" or "likely" to be affected by the pandemic, with a mean of 2.97 ($SD = 1.26$).

## General health and stressors

The mean of the sum score for PHQ-2 in our study was 2.32 which is below the cut-off of three for major depression [26]. Yet, 33.1% of the participants were above the cut-off. For the GAD-7, the sum score for the study's sample was 8.49. Cut points of 5 might be interpreted as mild, cut points of 10 as moderate and 15 as severe levels of anxiety [27], which implies a mild risk level for generalized anxiety with the suggestion of a follow-up examination in this sample. When asking for mental disorders, we found that 19.9% of the participants ($n = 99$) have already been diagnosed with a mental disorder and 15.5% ($n = 77$) are currently in psychotherapy. The sum score for the Perceived Stress Scale (PSS) of 7.79 (with *Min* = 0, *Max* = 15) was above the total sum score compared to a representative British sample (6.11) [36] and a representative German community sample (4.79 for PSS-4) [37]. Job satisfaction of our participants with a total sum score of 10.06 was lower compared to a sum score of 12.79 in a German sample of workers in small- and medium sized enterprises [38]. The mean score for job satisfaction was 3.35, also lower than in a sample of Ph.D. students in Belgium (3.9) [39]. Job insecurity was with a total sum score of 8.76 higher compared to the German small- and medium sized enterprises sample (5.67) [38]. Consistently, more than 80% of the Ph.D. students in our study were worried about the lack of permanent or long-term contracts in academia (*M* = 4.25, *SD* = 1.09; 5 indicating a strong agreement). Nevertheless, around half of the participants (54.5%) believed that having a Ph.D. would help them find a good job (*M* = 3.49, *SD* = 0.97). We found a mean score of 3.48 (*SD* = 0.98) for the positive support questions which is above average over response levels. Around 57.1% of the Ph.D. students felt supported by their supervisor "most" or "all of the time". Around 55.7% felt comfortable when contacting the supervisor for support. The negative support construct was with a mean score of 2.18 below average: 46.7% of the participants had never felt looked down, and 62.6% had never felt mistreated by their supervisor. Nevertheless, 28.6% of the Ph.D. students answered feelings of degradation and 19.1% felt mistreated more than "some of the time". When it comes to the frequency of the meetings with the supervisor, the mean reported a value of 2.4 laying somewhere between having meetings once a month (2) and at least every three months (3). However, 18.2% reported meeting their supervisor only once every six months or less. For sample items and detailed values see S2 Table.

When we analyzed the relationship between the studied outcomes, we found that all major constructs correlated significantly (see Table 1). High correlations occurred between the items of the related PHQ-2 and GAD-7 as well as their connections to the PSS. Understandably, the two institutional support dimensions were highly correlated ($r = -.69$).

## Regression for perceived stress, depression, and anxiety

To predict potential driving factors for the two more direct mental health measurements, namely depression and anxiety, and for perceived stress, we employed linear regression models with these three constructs as response variables controlling for age and gender. We included relevant risk factors and stressors such as job insecurity, perceived stress, negative support and resources such as job and life satisfaction, and positive support to get a comprehensible overview over predictors. All analyses were carried out in R statistics version 4.1.3.

For depression, significant predictors were job satisfaction ($\beta = -0.1$, *SE* = 0.04, $p < .05$), life satisfaction ($\beta = -0.3$, *SE* = 0.04, $p < .001$), perceived stress ($\beta = 0.4$, *SE* = 0.05, $p < .001$) and negative institutional support ($\beta = 0.11$, *SE* = 0.05, $p < .05$, see Table 2). The model explained 46.7% of the variance, $F(8, 482) = 54.5$, $p < .01$.

For anxiety, all studied variables except job satisfaction and positive support were significant predictors with a variance explanation of 36.0%, $F(8, 392) = 29.5$, $p < .01$ (see Table 2).

**Table 1. Correlations (Pearson *r*) for the constructs measuring general health, job insecurity, job, and life satisfaction as well as institutional and systemic stressors.**

| Variable | 1 | 2 | 3 | 4 | 5 | 6 | 7 |
|---|---|---|---|---|---|---|---|
| 1. Depression | | | | | | | |
| 2. Anxiety | .53** | | | | | | |
| | [.46, .60] | | | | | | |
| 3. Perceived stress | .59** | .52** | | | | | |
| | [.53, .65] | [.45, .59] | | | | | |
| 4. Job insecurity | .27** | .35** | .42** | | | | |
| | [.18, .35] | [.26, .43] | [.34, .49] | | | | |
| 5. Job satisfaction | -.44** | -.35** | -.43** | -.25** | | | |
| | [-.51, -.37] | [-.43, -.26] | [-.50, -.36] | [-.33, -.17] | | | |
| 6. Life satisfaction | -.61** | -.44** | -.58** | -.34** | .53** | | |
| | [-.66, -.55] | [-.52, -.36] | [-.63, -.52] | [-.42, -.26] | [.46, .59] | | |
| 7. Positive support | -.28** | -.27** | -.33** | -.25** | .54** | .36** | |
| | [-.35, -.19] | [-.36, -.18] | [-.41, -.25] | [-.33, -.17] | [.47, .60] | [.29, .44] | |
| 8. Negative support | .36** | .38** | .39** | .31** | -.57** | -.35** | -.69** |
| | [.28, .44] | [.30, .46] | [.31, .46] | [.23, .39] | [-.63, -.51] | [-.42, -.27] | [-.73, -.64] |

Depression = Perceived Health Questionnaire 2, Anxiety = Generalized Anxiety Disorder 7.

Values in square brackets indicate the 95% confidence interval for each correlation.

** $p < .01$.

Noticeable was the strong influence of perceived stress on anxiety. Specifically, we observed that with an increase of one unit in perceived stress, the level of GAD-7 increased by 2.02 units and was in line with the high correlation ($r = .52$, $p < .01$, Table 2).

For perceived stress, we found that job insecurity ($\beta = 0.15$, $SE = 0.02$, $p < .01$), life satisfaction ($\beta = -0.32$, $SE = 0.03$, $p < .01$) as well as negative institutional support ($\beta = 0.13$, $SE = 0.04$, $p < .01$) were significant predictors with a model variance explanation of 42.7%, $F(4, 486) = 53.5$, $p < .01$. The detailed results for this regression analysis can be found in S4 Table.

## Qualitative answers

In the following, we report the main categories with short sample quotes as well as the mean frequency of the two raters (see Table 3; details in S5–S7 Tables). The inter-rater reliability as indicated by Krippendorff's alpha for the top five categories of all questions was above $\alpha \geq .67$, except for the category *Manageable Workload* for question MH06_1 (see Table 3) with $\alpha = .62$; CI [0.50; 0.74]. A threshold of .67 is commonly considered as the lower conceivable limit that still allows tentative conclusions [40].

**Causes of stress.** The question "What is/are the cause(s) of your stress?" was answered by $n = 446$ participants. To cover the breadth of the responses, we built 18 categories. The most frequently mentioned categories were *Workload & Time Pressure* (mean rating frequency = 211), *Self-Perception* (M = 132.5), *Job-Insecurity* (M = 93), *Social Integration & Interactions* (M = 91), and *Supervision Quality & Quantity* (M = 88.5). The category *Workload & Time Pressure* includes all responses referring to the amount of work and/or deadlines. The category *Self-Perception* includes responses that indicate a perceived lack of competences or other personal doubts, concerns, and worries (e.g., "Since I started my Ph.D. I have almost constantly felt stupid", "feeling like not belonging in academia, lack of self-confidence, feeling of making too little progress"). The category *Job Insecurity* reflects responses regarding contract length and general uncertainty about future employment (e.g., "scholarship is to be

**Table 2. Linear regression models for depression (left) and anxiety (right).**

| Variable | Estimate | SE | p | Variable | Estimate | SE | p |
|---|---|---|---|---|---|---|---|
| PHQ-2 (Intercept) | 1.30 | 0.45 | | GAD-7 (Intercept) | -2.01 | 2.70 | |
| Age | -0.02 | 0.01 | | Age | 0.09 | 0.06 | |
| Gender | 0.01 | 0.06 | | Gender | -0.75 | 0.34 | * |
| Job insecurity | -0.01 | 0.03 | | Job insecurity | 0.42 | 0.17 | ** |
| Job satisfaction | -0.08 | 0.04 | * | Job satisfaction | -0.28 | 0.25 | |
| Life satisfaction | -0.30 | 0.04 | *** | Life satisfaction | -0.73 | 0.22 | ** |
| Perceived stress | 0.39 | 0.05 | *** | Perceived stress | 2.02 | 0.31 | *** |
| Positive support | 0.06 | 0.04 | | Positive support | 0.42 | 0.25 | |
| Negative support | 0.12 | 0.05 | * | Negative support | 1.01 | 0.29 | *** |

PHQ2 = Perceived Health Questionnaire; GAD-7 = Generalized Anxiety Disorder 7.

* $p < .05$

** $p < .01$

*** $p < .001$.

ended", "Not knowing how things will work out after the PhD", "Hopelessness of scientific career because there are too few full-time positions"). The category *Social Integration & Interactions* covers responses regarding the integration and sense of belonging in the work environment (e.g., "not valued by colleagues", "being socially isolated at work") as well as social issues in the private life (e.g., "Mostly my personal life, or often the lack thereof", "problems with parents"). The category *Supervision Quality & Quantity* was used to capture all supervision-related responses including comments about the lack of support, feedback, frequency of meetings, or supervisors' interest in the topics (e.g., "no clear communication with supervisor", "lack of support from supervisor, even gossiping about me behind my back").

**Potential ways to improve the mental health status.** When asked "What would need to change to improve your mental health status?", the Ph.D. students' responses ($n = 307$) included various topics, some addressing compensation and income-related aspects, others highlighting supportive supervision. Overall, the responses lead to twelve different categories. Most answers referred to *Supportive Supervision* ($M = 98.5$), followed by *Job Security/Contract* ($M = 59$). Sample quotes with respect to supervision are e.g., "more feedback from supervisor or even more interest in my topic" or "more regular support by supervisor". The category *Job Security/Contract* contains comments with respect to contract length and aspects for future employment (e.g., "no more worries about not being able to get my contract renewed"). The category *Manageable Workload* ($M = 56.5$) includes all responses around work-life balance (e.g., "having also activities beside work", "clear work hours"). The fourth category was *Compensation & Financial Security* ($M = 35$) and included all income- and compensation-related aspects of the job (e.g., "Be paid 100% would be a start", "Get paid for all the time at work"). The category *Less Additional Tasks* ($M = 27.5$) was used to specifically cover responses mentioning the number of additional tasks within the job ("Less work in teaching/work unrelated to PhD").

**Ways to improve the personal situation.** In addition to the previous question, which focused on general ways to improve the mental health status, we asked the Ph.D. students the following question: "What could be done to improve your situation?" Based on the themes and topics mentioned in the responses ($n = 281$) we built eleven categories. The categories mentioned the most were *Job-Security & Compensation* ($M = 85.5$), followed by *Supportive Supervision* ($M = 68$), *Services and Support System* ($M = 39.5$), *Decrease Pressure to Perform*

**Table 3. Results of qualitative content analyses.**

| Questions | Number of Categories | Top 5 Categories | Frequency Rater 1 | Frequency Rater 2 | Mean Rating Frequency | Kripp. α [95% CI] |
|---|---|---|---|---|---|---|
| *What is/are the cause(s) of your stress?* (MH03_01) *n* = 446 | 18 | 1. Workload & Time Pressure<br>2. Self-Perception<br>3. Job Insecurity<br>4. Social Integration & Interactions<br>5. Supervision: Quality & Quantity | 213<br>134<br>88<br>84<br>84 | 209<br>131<br>98<br>98<br>93 | 211<br>132.5<br>93<br>91<br>88.5 | .76 [0.70; 0.82]<br>.78 [0.72; 0.85]<br>.86 [0.80; 0.92]<br>.74 [0.66; 0.81]<br>.85 [0.79; 0.91] |
| *What would need to change to improve your mental health status?* (MH06_1) *n* = 307 | 12 | 1. Supportive Supervision<br>2. Job Security / Contract<br>3. Manageable Workload<br>4. Compensation & Financial Security<br>5. Less Additional Tasks | 95<br>65<br>63<br>41<br>30 | 102<br>53<br>50<br>29<br>25 | 98.5<br>59<br>56.5<br>35<br>27.5 | .77 [0.69; 0.84]<br>.69 [0.57; 0.78]<br>.62 [0.50; 0.74]<br>.74 [0.60; 0.86]<br>.78 [0.64; 0.90] |
| *What could be done to improve your situation? Feel free to express your opinion and feelings here.* (SH07_01) *n* = 281 | 11 | 1. Job-Security & Compensation<br>2. Supportive Supervision<br>3. Decrease Pressure to Perform<br>4. Services & Support System<br>5. Manageable Workload | 85<br>69<br>38<br>43<br>38 | 86<br>67<br>41<br>36<br>34 | 85.5<br>68<br>39.5<br>39.5<br>36 | .90 [0.84; 0.95]<br>.87 [0.79; 0.93]<br>.69 [0.54; 0.80]<br>.70 [0.56; 0.81]<br>.74 [0.60; 0.86] |

*n* = represents the total number of comments in this open answer field. The full list of categories can be found in S5–S7 Tables. The confidence intervals for Krippendorff's alpha are calculated with a bootstrap sample of 1000.

($M$ = 39.5), and *Manageable Workload* ($M$ = 36). The category *Job-Security & Compensation* includes responses like "chances of getting a long-term job in academia, not just the three-year programs" or "Fair payment (half of students get 50% others 65% even at the same institute)". For the category *Supportive Supervision* "Regular meetings with people who are supportive & have an expertise in my research topic" can serve as a sample quote. The category *Services and Support System* was built to cover the responses named a solution outside the working group and team, such as "it would be helpful to see a university-based psychologist outside of the regular working hours" or "more courses (or better communications about them) about stress management". The next category was labeled *Decrease Pressure to Perform* and included all responses that highlighted a high level of perceived pressure, such as "the performance pressure (every talk at a seminar is a job talk) is a big problem" or "Instead of pressuring academics to publish as much as possible, there should be more focus on the quality instead of the quantity of their articles/publication". The last category, *Manageable Workload*, contained answers with respect to the amount of work (e.g., "Normal working hours, having really free-time without having the feeling that I should be working, it should be normal to take all vacation days").

**Summary of the qualitative answers.** With respect to the open answers, it can be summarized that the factors named as causes for stress and the possible solutions cover a wide range of topics. However, there are reoccurring topics across all three questions, such as supervision,

workload, and job security. The role of supervision is a reemerging motif in the qualitative content analysis. While the quality and quantity of supervision were seen as a cause of stress, supportive supervision has a positive impact on the mental health status as well as the whole situation of the Ph.D. students. Furthermore, job insecurity was mentioned as an important stressor, while stable contracts and appropriate compensation for the work and fewer extra tasks were also added for improvement. Workload and time pressure were the most often stated causes of stress, followed by self-doubts and worries about not having enough competencies for the job. A manageable workload, fewer additional tasks, and a lower pressure to perform were indicated by the participants as valuable improvements.

## Discussion

### Summary of the main findings

The conducted survey investigates the mental health of Ph.D. students at a university in the southwest of Germany and gives insights into what causes stress and mental health disorders and where there is a need for further support services. Our qualitative and quantitative analyses revealed interesting and consistent results on the alarming situation of the mental health of Ph.D. students.

First, our quantitative results revealed that one-third of the participants were above the cut-off for depression which is an indicator of a high risk of depression that should be checked by a health professional. On average, the surveyed Ph.D. students were at a mild risk level for an anxiety disorder. While our study design does not allow us to diagnose mental illnesses, it identifies problems that need to be pursued further. It reveals some unhealthy working conditions and increased risks for mental illnesses. Our qualitative and quantitative results showed consistently that many of the most prominent issues for our study's participants are personal factors such as perceived stress, life satisfaction and self-doubt, but modulated by structural deficits such as financial and job security as well as workload and time pressure. The quantitative analyses revealed that life satisfaction, perceived stress and negative support are the main predictors for anxiety disorders as well as depression. Additionally, low job satisfaction was a significant predictor of depression and job insecurity for anxiety. Furthermore, we identified job insecurity, life satisfaction as well as negative institutional support as predictors for perceived stress.

Second and besides mental health problems, our quantitative analyses showed how supervision and the work environment played a role in the mental health and general well-being of Ph.D. students. Deficient supervision could affect Ph.D. students' perceived job insecurity and job dissatisfaction. Although good supervision was not a predictor for satisfaction, being comfortable with contacting the supervisor could lower the perceived stress. This shows the importance of the supervisor-student relationship and highlights the importance of the social work environment, which was also mentioned by study participants in the open-end questions. While the categories in the qualitative analyses mainly served to find recurring themes, they can also be used to distinguish between different levels. Some participants reflected causes of stress on a personal level (e.g., self-perception). In contrast, others set the focus on the supervisor level or working group level, or even on the more structural abstract level of the academic system.

Third, our study does not only investigate the mental health situation of Ph.D. students, but we also analyze how the situation and mental health status could be improved. Many suggestions were straightforward given the results of the causes of stress, i.e., bad supervision should be improved, and a secure income should be guaranteed. However, we were also able to show that Ph.D. students wish to make use of services and support systems that could be provided

by the university. Furthermore, less pressure to perform and a manageable workload with fewer additional tasks besides the Ph.D. project might decrease the stress level and improve mental health status.

Overall, detrimental mental health is a known problem in academia, and we show another example of its extent as well as opportunities for improvement at a German university.

## Comparison to other studies

Data on Ph.D. students' situation in Germany are scarce, and we, therefore, perform a broader comparison with Ph.D. students around the world. However, the results of this comparison should be taken with caution as our questionnaire and time of survey conduction are unique. We focus mainly on PHQ-2 [26] and GAD-7 [27], for which other studies in Germany during the pandemic showed that–compared to pre-COVID-19 reference values–these measurements were significantly increased [41]. Two studies conducted during the COVID-19 pandemic include the same scales [41,42] and reveal similar results for the general population in Germany, while in our later study from October to December 2021, the risk for anxiety and depression is slightly higher. In our study, one-third of the participants (33.1%) was above the cut-off for major depression, compared to the studies in a 1.5-year earlier timeframe, where 14.1% (March to May 2020; $n$ = 15704, 70.7% female gender; 42.6% university education) [42] and 21.4% (March to July 2020; $n$ = 16918; 69.7% female gender; 42.4% university education [41] of the participants with diverse occupations were above the cut-off. Furthermore, in our study, 39.2% of the participants were at the mild risk level for anxiety compared to 27.4% of the participants in an earlier study [41]. This shows the increase in depression and anxiety during the pandemic and even higher numbers in our study compared with the German general population. Nevertheless, compared to a survey at public research universities in the United States from May to July 2020, the number of doctoral students screened for major depressive disorder symptoms with the same measurements PHQ-2 was higher with 36% [43], indicating high numbers of mental issues in academia in several countries.

While using the same scales and items for job satisfaction and job insecurity, our study showed worse sum scores compared to a sample of employers and employees in small- and medium sized enterprises in Germany (December 2020 to May 2021; $n$ = 828; 53.7% female gender, $M$ = 41.5 years; 38.8% higher education entrance qualification) [38]. It seems that Ph. D. students have higher job insecurity and job dissatisfaction compared to workers in diverse branches and occupations. This may result from different contract types, as workers, especially in industrial sectors, have long-term contracts. The recurrent factor of time pressure and workload, also mentioned in the open-end questions, is backed up by the raw numbers of the contract types and working hours, which may also lead to job dissatisfaction. Although the mean contract type in our study is 63%, the mean number of hours dedicated to Ph.D. work ($M$ = 36.0, $SD$ = 15.6 hours) is almost in the range of a full-time position. What is more, the participants reported a total weekly workload ($M$ = 44.1, $SD$ = 11.4 hours) that exceeds a typical full-time position in Germany [44]. The discrepancy between Ph.D. work and corresponding contract types results in a mean of 12.1 hours of overwork per week (based on a 38.5-hour full-time contract, which is the standard contract for Ph.D. students in Germany). This is in line with previous studies where the authors found a mean of 12.6 hours of overwork per week for Ph.D. students in Science, Technology, Engineering, and Mathematics disciplines in Germany [45]. However, the authors did not include any further work obligations and corrected for contract types with low percentages, and thus the results are difficult to compare directly. Furthermore, we used gender as a control variable, which turned out to be statistically

significant for anxiety and stress. This is in line with related work where the female gender was reported to be higher correlated with mental disorders [2,17,46,47].

## Strengths and limitations

**Generalization.** While we aimed for our study to reflect the current situation for Ph.D. students as best as possible, there are points that are limiting the generalization of the results or are beyond the scope of this survey. First, we collected the data between October and December 2021, a time at which the ordinance on protection against risks of infection with the SARS-CoV-2 virus ("Coronavirus-Schutzverordnung") [48] was still in place in Germany and influenced private and working life. About one-third (33.5%) of our study population stated that it is very likely or likely that the pandemic affected their answers. Nonetheless, a pandemic is a situation that can reoccur and is only one more reason to proactively set up a resilient Ph.D. graduation system. Another research group [49] investigated how mental health care should change as a consequence of the COVID-19 pandemic and concluded that the pandemic could even be seen as a chance to improve mental health services [49]. Nevertheless, we would like to point out that generalizing from a mental health study conducted during a pandemic may be difficult.

Overall, around 23% of all Ph.D. students at the University of Tübingen [50] participated in our study, which is slightly below the response rate in other similar studies [e.g., 16]. Considering that university students are very frequently invited to various questionnaires and studies, and given that our survey lasted approximately 20 minutes, it can be argued that the participants were motivated to invest time into their responses. However, our study population remains small compared to the total number of Ph.D. students in Germany. Moreover, we want to emphasize the likely sample bias in our data. We recruited participants mainly via mailing lists and our project therefore probably has especially appealed to people who are already interested in health or aware of mental health issues. However, given our relatively large coverage of almost a quarter of all Ph.D. students at the University of Tübingen, even a selective sample can give us insights into overall tendencies. The transferability of our results to other German universities or even universities in other countries is also not guaranteed as the academic systems can largely differ. Additionally, the results of this study are influenced by the overall living conditions the Ph.D. students experience. As Tübingen is a small town in the southwest of Germany, a comparison to larger cities or other countries might not be viable as the conditions probably differ largely.

Finally, even within one university, the generalization of our results is further limited by the uneven distribution of the participants across faculties. Most participants (61.8%) were from the Science Faculty, which is also the largest department (in terms of the highest total number of students) at the University of Tübingen. This skewness limits the faculty-wise comparisons, and we would expect to find interesting insights into the different graduate programs by conducting detailed comparisons. These differences could not only arise from different academic traditions but also from the highly varying expectations on the scope of a Ph.D. thesis. It follows that more detailed and systematic monitoring and data collection in national and international surveys are needed.

**Methodology.** In a cross-sectional study, we investigate the current situation of Ph.D. students. While this is a valid and important instrument to access the current state, it cannot give us information about the dynamic changes in the transition phase between undergraduate studies and the Ph.D. as well as across the Ph.D. [51]. To track these changes or make comparisons over time, a longitudinal study design or propensity score matching procedures [52] could give further insights. It is therefore desirable to establish regular surveys and monitoring systems either on a university level or in a national survey to provide information on the

impact of undertaken actions and implemented changes. We used a mixed quantitative and qualitative research approach. While this provides information on distinct levels, there are some pitfalls. For example, the open answer categories were defined post-hoc. While this gives the possibility for the participants to express their thoughts freely, it makes a systematic analysis more difficult, and the analysis might be biased by the evaluators. Overall, it is important to summarize and statistically analyze our study results on an overall level, but it must not be forgotten that every person and Ph.D. project is individual.

## Implications for research and practice

**Research.**   The overall scarce data, paired with worrisome flashlights on the mental health situation of Ph.D. students in different countries, highlights the need for more systematic monitoring of mental health in academia. For this purpose, standardized as well as domain-specific scales for Ph.D. students need to be established and longitudinal data needs to be collected. This would enable researchers to measure the effect of larger environmental changes (such as the COVID-19 pandemic or economic developments) and to measure the impact of interventions targeted to improve the situation. At the same time, we propose including qualitative measurements to assess unknown variables and the unique situation each Ph.D. student faces. These could also inform the development of additional quantitative measurable constructs to reflect the dynamic situation in academia. Such monitoring systems can either be implemented at the university level to give detailed insights into the situation at a specific university or on a national level to get an overall impression of Ph.D. students' health issues. Optimally, a survey should be promoted from an independent self-governing institution dedicated to advancing science and research. While the demands for a better mental health situation for Ph.D. students are obvious, systematical and political changes need to be addressed in the research community and in academia.

**Practice.**   Our mixed methods research approach allows us not only to find out more about the issues of Ph.D. students but also to draw conclusions about what is needed to improve their situation. However, finding solutions to a recognized problem is not a straightforward task, and complex problems often require a step-by-step solution. Therefore, we assume that more practical implications, which could be indicated by an established monitoring system, will be necessary once the first steps have been taken.

In general, we can group interventions into at least four levels that can influence each other: the Ph.D. students themselves, the supervisors, the universities or research institutions, and the greater political context and academic culture. Building on the responses about potential improvements and additional services, we identified the following practical implications:

On an individual level, the main interventions could happen in capacity building (e.g., in time/project management, self-reflection or mental health awareness) but also by being more proactive about changing working modes (e.g., establishing collaborations or a peer counseling system) or by improving the social environment. This could additionally lead to a change in self-perception, for which direct interventions might be more difficult. At this point, we want to highlight that changes on the individual level aim to prevent the development of mental health problems and strengthen the resilience of Ph.D. students. They can at no point replace professional support once such problems have been manifested.

The level of supervision seems to be the most urgent and promising target for an improvement of Ph.D. students' situation. As supervisors are usually defining a project and its goals, but also additional teaching or other tasks, they are responsible for setting the workload and time constraints. Not only the hard constraints of the working conditions but also the quality of supervision was often mentioned to be highly deficient. Possible interventions could target

improving the skills in personnel management of supervisors. But also, clear supervision requirements and guidelines could be imposed by the university. Such agreements (including expectations on the thesis, supervision times and conciliation mechanisms) might be an option to enhance the agreements in a supervisor-student relationship. While these suggestions are not new, and some of them are theoretically established in some university departments, our study results suggest that they are often ignored or not properly implemented, and more binding agreements and control mechanisms need to be made. Establishing additional external supervision, where for example the personnel management is reflected, might also give new perspectives and enhance demanding situations. At this point, it has to be considered that there are strong dependencies between Ph.D. students and their supervisors since, in many cases, it is the supervisors who have a major impact on the outcome of a Ph.D. thesis, such as the final grade. It remains challenging how Ph.D. students can criticize the supervising situation without negatively impacting the personal relationship with their supervisors.

Further interventions on the level of universities and research institutions might include support in bureaucratic processes and providing more information on different contact points (e.g., for mental health services). It is obvious that the aforementioned interventions (such as capacity building courses for Ph.D. students and supervisors) are dependent on the support of the central facilities of the research institution. Furthermore, highlighting the high prevalence of mental health problems, for example, at mandatory introductory sessions for Ph.D. students, might help to raise awareness about this topic. This could help unexperienced young researchers to notice signs of anxiety and depression early on before these mental disorders manifest. Finally, public events on this topic could reduce the stigma associated with it, making it easier for affected Ph.D. students to seek help. Such events might also be used to remind the students that it is important to take care not only of their physical but also mental health, for instance, by strengthening social relationships and pursuing hobbies which are not work-related.

Lastly, there are also changes in the political setting and academic culture needed. This includes a fair payment system, reasonable control of contract lengths and extensions, and more perspectives for long-term positions in academia. Considering that the vast majority of Ph.D. students will end up in positions outside of academia, it could be beneficial to better prepare students for careers in alternative job markets, such as industry. Such interventions might directly influence the job insecurity and job dissatisfaction of Ph.D. students. In Germany, the current regulations for temporary academic employment are being evaluated [53], but even propositions from the conference of university rectors [54] seem not to be sufficient for fundamental changes. These changes would also need a shift in the academic culture [55], in which "publish or perish" is still a guiding theme leading to high pressure to perform. Working on a cultural shift is a task for all scientists. This will lead to a more sustainable work culture from which all stakeholders might benefit.

All in all, there is an interplay and dependence of all mentioned levels. Importantly, most problems mentioned in the survey can result from shortcomings on multiple levels, and therefore interventions on more than one level are needed for a satisfying solution. For example, changes to improve the mental health situation on an individual level can be dependent on the consent of the supervisor and can also be negatively impacted by already existing mental health issues. In addition to individual responsibility for health, it is important to systematically target prevention and change the system on the aforementioned levels so that Ph.D. students are better and more quickly supported when mental health problems arise.

## Conclusion

This study shows once again the detrimental mental health situation of Ph.D. students in academia. By analyzing the mental health of Ph.D. students at a German university, we found alarming hints of depressive and anxious tendencies that are in line with other comparable studies. Furthermore, we have identified main stressors, such as perceived stress or self-doubts, and resources, such as a positive student-supervisor relationship. Understanding conditional factors and being able to improve the situation depend on such identifications. With our study, we provide first insights of the status quo for the University chair, the Graduate Academy, and other stakeholders in the academic system. We invite them to inspect the results and suggestions responsibly so that actions to assess and improve the conditions for Ph.D. students' mental health and well-being can be taken in the future. Based on our data, additional offers for Ph.D. students, as well as their supervisors, should be created and existing ones sustainably modified. Positive conditions and resources for mental health and well-being will not restrict to academia but will affect all areas of life. While an increased mental health state is an indispensable value on its own, additional benefits can be created for research, teaching, practice, and society. As such, mental health is a big part of sustainable living and should have a high priority for all people. While this is already acknowledged in the sustainable development goals, further steps need to be taken to raise awareness and provide support throughout society.

## Supporting information

**S1 Table. Sample items and descriptives of Ph.D. students (*n* = 589): Percentage (%), mean (*M*), standard deviation (*SD*), minimum and maximum (*Min-Max*).**
(DOCX)

**S2 Table. Used scales and items with percentage (%), mean (*M*), standard deviation (*SD*), minimum and maximum (*Min-Max*), *median*, Cronbach's *alpha*.**
(DOCX)

**S3 Table. Faculty wise mean comparison on the job insecurity scale.**
(DOCX)

**S4 Table. Linear regression model for perceived stress and the predictors.**
(DOCX)

**S5 Table. Categories and ratings for the causes of stress.**
(DOCX)

**S6 Table. Categories and ratings for an improvement of mental health.**
(DOCX)

**S7 Table. Categories and ratings for an improvement of the situation.**
(DOCX)

## Acknowledgments

We would like to express our gratitude to all participants of the survey as well to the *sustainAbility* Ph.D. initiative at the University of Tübingen. We thank Dr. Stephanie Rosenstiel for support with the ethics approval and Prof. Dr. Birgit Derntl and Prof. Dr. Andreas Fallgatter for their helpful feedback on the conception of the questionnaire. We thank Mumina Javed and Monja Neuser for their support in the early phase of the project.

## Author Contributions

**Conceptualization:** Julian Friedrich, Anna Bareis, Moritz Bross, Zoé Bürger, Álvaro Cortés Rodríguez, Nina Effenberger, Markus Kleinhansl, Cornelius Schröder.

**Data curation:** Julian Friedrich, Anna Bareis, Moritz Bross, Álvaro Cortés Rodríguez, Nina Effenberger, Markus Kleinhansl, Cornelius Schröder.

**Formal analysis:** Julian Friedrich, Álvaro Cortés Rodríguez, Nina Effenberger, Markus Kleinhansl, Cornelius Schröder.

**Investigation:** Julian Friedrich.

**Methodology:** Julian Friedrich, Álvaro Cortés Rodríguez, Nina Effenberger, Markus Kleinhansl, Cornelius Schröder.

**Project administration:** Julian Friedrich.

**Software:** Cornelius Schröder.

**Supervision:** Julian Friedrich, Cornelius Schröder.

**Writing – original draft:** Julian Friedrich, Anna Bareis, Álvaro Cortés Rodríguez, Nina Effenberger, Markus Kleinhansl, Cornelius Schröder.

**Writing – review & editing:** Julian Friedrich, Anna Bareis, Moritz Bross, Zoé Bürger, Álvaro Cortés Rodríguez, Nina Effenberger, Markus Kleinhansl, Fabienne Kremer, Cornelius Schröder.

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
