## [Decision Letter · Decision Letter 0]

4 May 2023

PONE-D-23-08552“How is your thesis going?” – Ph.D. students’ perspectives on mental health and stress in academiaPLOS ONE

Dear Dr. Friedrich,

Thank you for submitting your manuscript to PLOS ONE. After careful consideration, we feel that it has merit but does not fully meet PLOS ONE’s publication criteria as it currently stands. Therefore, we invite you to submit a revised version of the manuscript that addresses the points raised during the review process.

Thank you for submitting this study. Please conduct the minor changes suggested by our reviewers. "Comments: 1. The topic is extremely interesting and relevant. It seems to be a well-developed study. 2. I suggest using controlled vocabulary terms for the Key words (Mesh) section. 3. As mentioned by the authors, there are limitations associated to the study design they are developing (prevalence). In the discussion section, I would like the authors to deepen on others study designs (maybe more appropriate) to would allow handling confounding factors. 4. Given the large sample size, I would like to know if they performed any calibration of the tool and if a pilot phase was conducted. 2. Did you receive any assistance in preparing this review (e.g. from a post-doc or graduate student)? If yes, please include their name below

We look forward to receiving your revised manuscript.

Kind regards,

Khader Ahmad Almhdawi, Ph.D

Academic Editor

PLOS ONE

Journal Requirements:

"We acknowledge support by the Open Access Publishing Fund of University of Tübingen."

"We would like to express our gratitude to all participants of the survey as well to the sustainAbility Ph.D. initiative at the University of Tübingen. We thank Dr. Stephanie Rosenstiel for support with the ethics approval and Prof. Dr. Birgit Derntl and Prof. Dr. Andreas Fallgatter for their helpful feedback on the conception of the questionnaire. We thank Mumina Javed and Monja Neuser for their support in the early phase of the project. We acknowledge support by the Open Access Publishing Fund of the University of Tübingen."

"We acknowledge support by the Open Access Publishing Fund of University of Tübingen."

"The authors have declared that no competing interests exist. All authors are or were Ph.D. students at the University of Tübingen."

7. Your ethics statement should only appear in the Methods section of your manuscript. If your ethics statement is written in any section besides the Methods, please move it to the Methods section and delete it from any other section. Please ensure that your ethics statement is included in your manuscript, as the ethics statement entered into the online submission form will not be published alongside your manuscript. 

Additional Editor Comments:

Thank you for submitting this study. Please conduct the minor changes suggested by our reviewers.

"Comments:

1. The topic is extremely interesting and relevant. It seems to be a well-developed study.

2. I suggest using controlled vocabulary terms for the Key words (Mesh) section.

3. As mentioned by the authors, there are limitations associated to the study design they are developing (prevalence). In the discussion section, I would like the authors to deepen on others study designs (maybe more appropriate) to would allow handling confounding factors.

4. Given the large sample size, I would like to know if they performed any calibration of the tool and if a pilot phase was conducted.

2. Did you receive any assistance in preparing this review (e.g. from a post-doc or graduate student)? If yes, please include their name below"

Reviewers' comments:

Reviewer's Responses to Questions

**Comments to the Author**

1. Is the manuscript technically sound, and do the data support the conclusions?

Reviewer #1: Yes

Reviewer #2: Yes

2. Has the statistical analysis been performed appropriately and rigorously? 

Reviewer #1: Yes

Reviewer #2: Yes

3. Have the authors made all data underlying the findings in their manuscript fully available?

Reviewer #1: Yes

Reviewer #2: Yes

4. Is the manuscript presented in an intelligible fashion and written in standard English?

Reviewer #1: Yes

Reviewer #2: Yes

5. Review Comments to the Author

Reviewer #1: 1. The topic is extremely interesting and relevant. It seems to be a well-developed study.

2. I suggest using controlled vocabulary terms for the Key words (Mesh) section.

3. As mentioned by the authors, there are limitations associated to the study design they are developing (prevalence). In the discussion section, I would like the authors to deepen on others study designs (maybe more appropiate) to would allow handling confounding factors.

4. Given the large sample size, I would like to know if they performed any calibration of the tool and if a pilot phase was conducted.

Reviewer #2: Dear Authors,

I thoroughly enjoyed reading your paper. This paper has full merit in getting accepted in Plos One for publication. However, I found some small mistakes in the manuscripts, such as the parenthesis needing to be closed properly in some places.

I recommend this manuscript for publication in Plos One.

Thanks

6. PLOS authors have the option to publish the peer review history of their article (what does this mean?). If published, this will include your full peer review and any attached files.

Reviewer #1: **Yes: **KT DIAZ

Reviewer #2: **Yes: **Dinakaran Elango

---

## [Author Response · Author response to Decision Letter 0]

7 Jun 2023

Dear Khader Ahmad Almhdawi,

Dear Reviewers KT DIAZ and Dinakaran Elango,

We thank you for your feedback and insightful comments and the opportunity to improve our manuscript. We want to highlight that both reviewers were extremely positive about the presented work and “enjoyed reading” our manuscript and described the topic as “extremely interesting and relevant”. We really appreciate your enthusiasm and want to thank you for your time and energy to sharing your expertise with us. We have thoroughly answered each of the comments and provide a point-by-point response below.

Reviewer 1

1. The topic is extremely interesting and relevant. It seems to be a well-developed study.

Response: We thank the reviewer for these positive comments. 

2. I suggest using controlled vocabulary terms for the Key words (Mesh) section.

Response: 

Thanks for this great suggestion. We investigated the Medical Subject Headings on https://meshb.nlm.nih.gov and changed the keywords to controlled terms:

“mental health, depression, working conditions”. Additionally, we included specific terms for our context (“Ph.D. students”, “academia”), which we think are very informative but don’t have a proper correspondence in the aforementioned keyword mesh.

3. As mentioned by the authors, there are limitations associated to the study design they are developing (prevalence). In the discussion section, I would like the authors to deepen on others study designs (maybe more appropriate) to would allow handling confounding factors.

Response: 

We thank the reviewer to highlight the need to mention other study designs. In the updated version of the manuscript, we now discuss “about the dynamic changes in the transition phase between undergraduate studies and the Ph.D. as well as across the Ph.D. [51]. To track these changes or make comparisons over time, a longitudinal study design or propensity score matching procedures [52] could give further insights.” Other methods could be difference-in-difference approaches or regression discontinuity designs.

4. Given the large sample size, I would like to know if they performed any calibration of the tool and if a pilot phase was conducted.

Response: 

We conducted an extensive literature review for the development of the questionnaire and identified relevant constructs for academic and work contexts. We discussed the questionnaire with two professors (with psychological and medical background) and iteratively developed it further. Subsequently, we tested the questionnaire several times within the research team and with members of the sustainAbility Ph.D. initiative for timing and comprehensibility, including interdisciplinary perspectives. However, a systematic pilot phase was not conducted.

2. Did you receive any assistance in preparing this review (e.g. from a post-doc or graduate student)? If yes, please include their name below."

Response:

We did not receive any other assistance apart from the ones named in the acknowledgments: For the ethics approval, we received support from Dr. Stephanie Rosenstiel. For the questionnaire development, we consulted with Prof. Dr. Andreas Fallgatter and Prof. Dr. Brigit Derntl. 

Reviewer 2

Dear Authors,

I thoroughly enjoyed reading your paper. This paper has full merit in getting accepted in Plos One for publication. However, I found some small mistakes in the manuscripts, such as the parenthesis needing to be closed properly in some places.

I recommend this manuscript for publication in Plos One. 

Thanks

Response:

We thank the reviewer for the positive feedback. We checked again thoroughly the manuscript for spelling and parentheses. The changes are highlighted in the appended version.

Editor

Response: 

We thank the editor for this reminder. We changed the manuscript accordingly and hope that it fulfils all requirements. 

Response: 

We thank the editor for this comment. We updated the grant information in the “funding information” and “financial disclosure”. There are no grant numbers for the open access fund (“We acknowledge support by the Open Access Publishing Fund of the University of Tübingen.”).

"We acknowledge support by the Open Access Publishing Fund of University of Tübingen."

Response: 

We thank the editor for this comment. We would like to add the provided suggestion: “We acknowledge support by the Open Access Publishing Fund of University of Tübingen. The funders had no role in study design, data collection and analysis, decision to publish, or preparation of the manuscript.”

"We would like to express our gratitude to all participants of the survey as well to the sustainAbility Ph.D. initiative at the University of Tübingen. We thank Dr. Stephanie Rosenstiel for support with the ethics approval and Prof. Dr. Birgit Derntl and Prof. Dr. Andreas Fallgatter for their helpful feedback on the conception of the questionnaire. We thank Mumina Javed and Monja Neuser for their support in the early phase of the project. We acknowledge support by the Open Access Publishing Fund of the University of Tübingen."

"We acknowledge support by the Open Access Publishing Fund of University of Tübingen."

Response: 

We thank the editor for this clarification. We deleted the funding information in the acknowledgements. The funding statement can remain as it is: “We acknowledge support by the Open Access Publishing Fund of University of Tübingen. The funders had no role in study design, data collection and analysis, decision to publish, or preparation of the manuscript.”

"The authors have declared that no competing interests exist. All authors are or were Ph.D. students at the University of Tübingen."

Response: 

We thank the editor for this clarification. We would like to change the sentence to “The authors have declared that no competing interests exist.” and delete the next sentence “All authors are or were Ph.D. students at the University of Tübingen.", as this information is already provided in the authors’ affiliation list. 

Response:

We thank the editor for this comment and highly appreciate the initiative for freely accessible research data. The minimal data set is now available online. The following link is also the DOI: https://doi.org/10.23668/psycharchives.12914. The code is provided in a public GitHub repository: https://github.com/coschroeder/mental_health_analysis.

The updated Data Availability statement reads as follows: “The anonymized data set is available at https://doi.org/10.23668/psycharchives.12914. All code for the analysis can be found at https://github.com/coschroeder/mental_health_analysis.”

7. Your ethics statement should only appear in the Methods section of your manuscript. If your ethics statement is written in any section besides the Methods, please move it to the Methods section and delete it from any other section. Please ensure that your ethics statement is included in your manuscript, as the ethics statement entered into the online submission form will not be published alongside your manuscript.

Response: 

We thank the editor for this comment. The ethics statement was already mentioned in the Methods section on p. 5: “Ethics approval was obtained by the “Ethics Committee of the Faculty of Economics and Social Science of the University of Tübingen”, which did not raise any ethical concerns. Written informed consent was given by the participants.” We deleted it from other sections.

Response: 

We thank the editor for this comment. We reviewed and updated our reference list. We did not cite any retracted papers. All changes are highlighted in the attached document.

Sincerely,

Julian Friedrich and the authors

julian.friedrich@uni-tuebingen.de

---

## [Editor Report · Decision Letter 1]

20 Jun 2023

“How is your thesis going?” – Ph.D. students’ perspectives on mental health and stress in academia

PONE-D-23-08552R1

Dear Dr. Friedrich,

We’re pleased to inform you that your manuscript has been judged scientifically suitable for publication and will be formally accepted for publication once it meets all outstanding technical requirements.

Kind regards,

Khader Ahmad Almhdawi, Ph.D

Academic Editor

PLOS ONE

Additional Editor Comments (optional):

Thank you for adopting reviewers' comments and improving your manuscript. We think it is ready for publication, congratulations!
---

## [Editor Report · Acceptance letter]

23 Jun 2023

PONE-D-23-08552R1 

“How is your thesis going?” – Ph.D. students’ perspectives on mental health and stress in academia 

Dear Dr. Friedrich:

I'm pleased to inform you that your manuscript has been deemed suitable for publication in PLOS ONE. Congratulations! Your manuscript is now with our production department. 

Kind regards, 

on behalf of

Prof. Khader Ahmad Almhdawi 

Academic Editor

PLOS ONE